ecology, evolution, neuroscience

anura, diel pattern, ocular media transmittance, ultraviolet sensitivity, vision, visual ecology

**Author for correspondence:**
Carola A. M. Yovanovich
e-mail: carola.yovanovich@ib.usp.br

# Lens transmittance shapes ultraviolet sensitivity in the eyes of frogs from diverse ecological and phylogenetic backgrounds

Carola A. M. Yovanovich[1], Michele E. R. Pierotti[1,2], Almut Kelber[3], Gabriel Jorgewich-Cohen[1], Roberto Ibáñez[2] and Taran Grant[1]

[1]Department of Zoology, Institute of Biosciences, University of São Paulo, São Paulo, Brazil
[2]Smithsonian Tropical Research Institute, Panama City, Panama
[3]Department of Biology, Lund University, Lund, Sweden

CAMY, 0000-0002-0589-3499; MERP, 0000-0003-2837-6192; AK, 0000-0003-3937-2808; GJ-C, 0000-0001-9807-6297; TG, 0000-0003-1726-999X

The amount of short wavelength (ultraviolet (UV), violet and blue) light that reaches the retina depends on the transmittance properties of the ocular media, especially the lens, and varies greatly across species in all vertebrate groups studied previously. We measured the lens transmittance in 32 anuran amphibians with different habits, geographical distributions and phylogenetic positions and used them together with eye size and pupil shape to evaluate the relationship with diel activity pattern, elevation and latitude. We found an unusually high lens UV transmittance in the most basal species, and a cut-off range that extends into the visible spectrum for the rest of the sample, with lenses even absorbing violet light in some diurnal species. However, other diurnal frogs had lenses that transmit UV light like the nocturnal species. This unclear pattern in the segregation of ocular media transmittance and diel activity is shared with other vertebrates and is consistent with the absence of significant correlations in our statistical analyses. Although we did not detect a significant phylogenetic effect, closely related species tend to have similar transmittances, irrespective of whether they share the same diel pattern or not, suggesting that anuran ocular media transmittance properties might be related to phylogeny.

## 1. Background

The lenses of animal eyes need to be transparent to allow the light they focus to reach its final destination, the photoreceptors in the retina. Among vertebrates, this condition is met invariably in the range approximately 450–700 nm, but there is great variability across the ultraviolet (UV)–blue range of the spectrum (approx. 300–450 nm) [1]. Part of this variation is owing to the chemical composition of the lens, as the proteins that comprise it absorb light strongly below 310 nm [1]. Thus, the bigger the lens, the greater the distance the light must traverse and the higher the probability that short-wavelength light will be absorbed. Furthermore, light is also scattered by particles in the ocular media generally and the lens specifically, although this scattering is thought to be wavelength-independent [1]. A partial reduction in UV transmittance is, thus, a consequence of eye enlargement, which, in turn, enables high sensitivity and better spatial resolution [2]. In addition to this trade off, there are optical benefits to modulating the spectral composition of light available for the retina, as short wavelengths are particularly prone to another type of scattering caused by small particles (Rayleigh scattering) and also to chromatic aberrations [1,3]. Prolonged exposition to short-wavelength light can also cause photochemical damage to the retina [4], so it has been proposed that long-lived and diurnal animals might benefit from UV-absorbing lenses [5].

A widespread solution to the problems caused by short-wavelength light is to add pigments to the lens to filter it. Such pigments have been spectrally and chemically characterized in some fishes [6,7], mammals [5] and the leopard frog [7] and inferred to exist in some birds [8]. A different strategy to cope with chromatic aberrations without filtering light are multifocal lenses, which allow light of different wavelengths to converge on the same focal plane via a specific refractive index gradient [9]. This mechanism obviously requires the whole lens to be exposed to the incoming light and would not work properly in eyes with round pupils that cover the periphery of the lens when contracted. Indeed, multifocal lenses strongly correlate with pupil shape in vertebrates: in a sample of 20 species from different tetrapod groups, all the species that have slit-shaped pupils also have multifocal lenses [10]. Thus, a combination of a highly short-wavelength transmissive lens with multifocal optics and a slit pupil can be an alternative to a pigmented, short-wavelength absorbing lens if photoreceptors have sensitivity peaks at short wavelengths or if light availability needs to be maximized, even at the cost of some scattering.

Lifestyle and geographical distribution determine the amount and spectral composition of light to which animals are exposed, both in the temporal (day, night) and spatial dimensions (latitude, elevation and habitat type [11–13]), and, in turn, the lens (and occasionally the cornea) can selectively filter part of that light. Thus, it is reasonable to expect that lens transmittance properties would have evolved in such a way that they 'match' the light environment in which a given visual system performs. Accordingly, it has been hypothesized [14] that nocturnal animals would have highly transmissive lenses to maximize the number of photons that can reach the retina in a context in which they are scarce per se, and that diurnal animals for whom light is an 'unlimited' resource could afford to filter out part of the short-wavelength radiation to fine tune resolution while preventing the potential damage caused by the exposure to high amounts of that kind of radiation. Indeed, there seems to be a loose tendency for this to occur in fishes [6,15,16], snakes [17] and mammals [5], with exceptions in all cases, although the relationship between lens transmittance and diel pattern has not been statistically tested in any of them. Several studies have investigated the variability in lens transmittance at short wavelengths and its correlation with eye size, photoreceptor spectral sensitivity and a variety of natural history traits in fishes [16,18], lizards [19], snakes [17], birds [8,20] and mammals [5]. However, to our knowledge, no broad comparative study of ocular media transmittance in amphibians has been pursued so far, and none of the studies in other lineages quantified those relationships in a formal phylogenetic context, so the interplay between ecology and evolutionary history in shaping the light transmittance properties in vertebrate eyes remains virtually unexplored.

In a previous study, we showed that the lenses of two species of anurans widely used as experimental models in vision research differ by more than 50 nm in the cut-off wavelength at which 50% of incoming light is transmitted ($\lambda_{T50}$) [21]. Although that study also included another three closely related species, a broader sampling was clearly needed to unveil the variability of lens transmittance across anuran species and lineages, and to assess potential relationships with their natural histories and with other properties of the visual system. In the present study, we assessed the lens transmittance, eye size and pupil morphology of 37 species sampled from across the diversity of anurans and evaluated their relationship with the temporal and geographical environments that they inhabit.

## 2. Methods

### (a) Sampling

We used eyes from 32 species of neobatrachian amphibians collected in their natural habitats in Brazil and Panama and one captive specimen of the basal species *Bombina orientalis* that died for reasons unrelated to the study (see figure 1 for taxonomic distribution and phylogenetic relationships and electronic supplementary material, S1A for details on identity and provenance of all specimens). We euthanized all the other animals by topical application of 20% benzocaine on the ventral skin or by partial immersion in $2\,g\,l^{-1}$ tricaine methanesulfonate (MS 222) buffered at pH 7 with sodium bicarbonate, until their breathing and cardiac activity ceased. In all cases, after death/euthanasia, we enucleated the eye, freed the cornea by cutting along the *ora serrata*, cut through the vitreous to extract the lens and removed the iris by cutting through the aqueous humour to obtain isolated corneas and lenses. All samples were measured immediately after dissection.

### (b) Data collection

We measured lens transmittance (and corneal transmittance for some species) using the approach of Lind *et al*. [20], as follows. We placed the samples in a custom-made matte black plastic container with a circular fused silica window in the bottom and filled with 0.01 M phosphate-buffered saline (PBS). For small samples, black plastic discs with pinholes of 1 or 2 mm diameter were added on top of the silica window to ensure that all incoming light passed through the sample. We used an HPX-2000 Xenon lamp (Ocean Optics, Dunedin, FL, USA) to illuminate the samples via a 50 μm light guide (Ocean Optics) and collected transmitted light using a 1000 μm guide connected to a Maya2000 spectroradiometer controlled by SPECTRASUITE v. 4.1 software (Ocean Optics). The guides were aligned with the container in a microbench system (LINOS, Munich, Germany). The reference measurement was taken from the container filled with PBS. We smoothed the curves using an 11-point running average, and normalized to the highest value within the range 300–700 nm. From these data, we determined $\lambda_{T50}$ as the wavelength at which the light transmittance was 50% of the maximum. The curves were cut to avoid clutter in those cases in which the measurements at very low wavelengths were too noisy owing to the low sensitivity of the spectrometer in that region of the spectrum.

We combined the lens transmittance data collected from the 32 species measured in this study with those from *Bufo bufo*, *Rhinella ornata*, *Lithobates catesbeianus*, *Lithobetes pipiens* and *Rana temporaria* that were available from a previous study [21], making a total of 37 species of 14 families. Given that corneal transmittance data were collected from just a handful of species, they were not included in the phylogenetic comparative analyses.

We used eye size compiled from descriptions of the species in the scientific literature as a proxy for lens optical path length. When these data were not available for a given species, we obtained them from colleagues or measured it from museum specimens (see the electronic supplementary material, S1B for the whole dataset of eye size values and sources, and the electronic supplementary material, S1C for validation of the method).

For pupil shape, we visually inspected photographs available online for each of the species and scored them as round or

Proc. R. Soc. B 287: 20192253

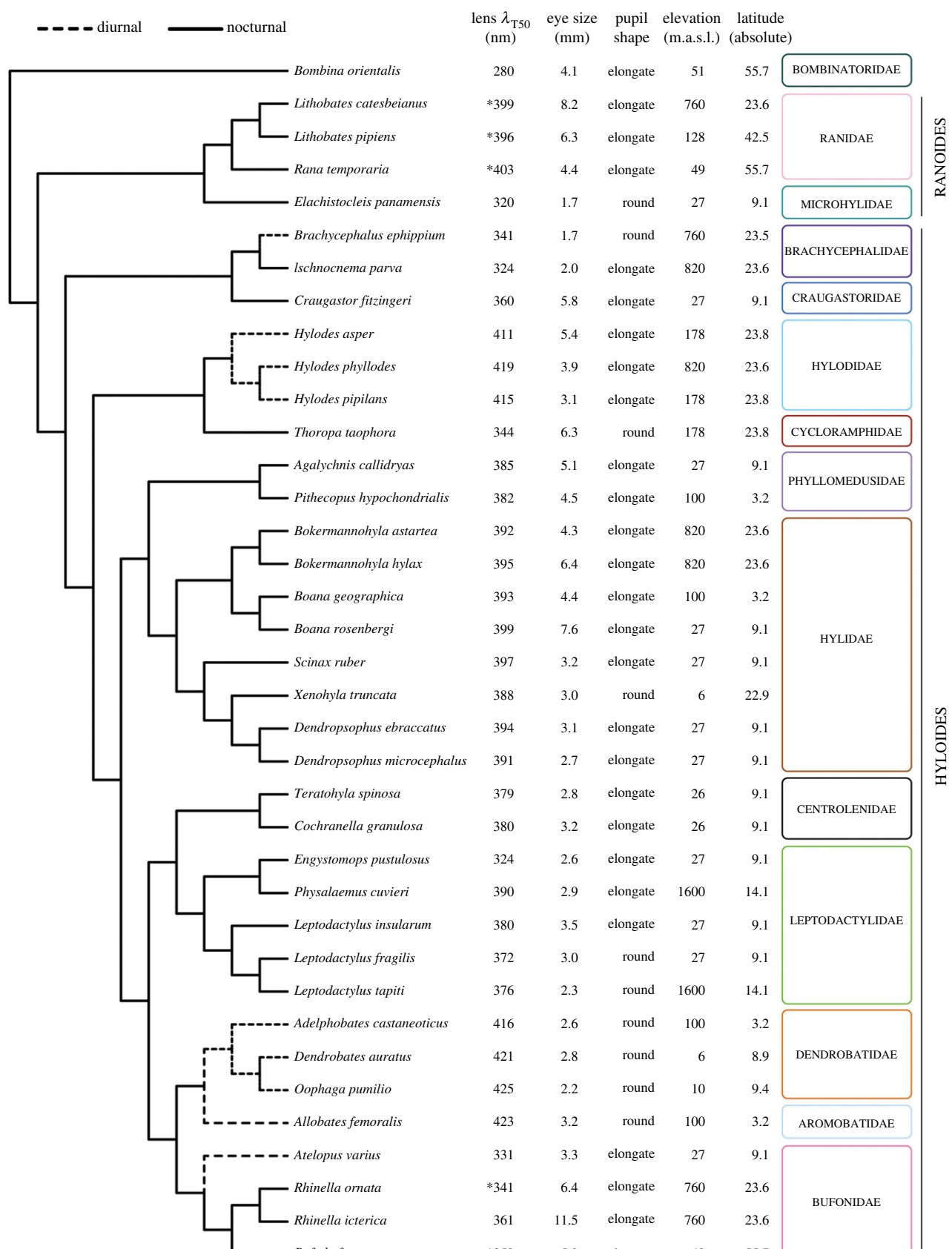

**Figure 1.** Summary of the values obtained for all the variables measured and compiled for the frog species included in this study in a phylogenetic context (see the electronic supplementary material, S1F for the tree with branches lengths scaled to phylogenetic distances). All the lens $\lambda_{T50}$ were calculated in this study except those marked with an asterisk (*), which were obtained from [21]. Full lens transmittance datasets are available in the electronic supplementary material, S2.

elongate (see the electronic supplementary material, S1D for details and thresholds on scoring criteria). Even though orientation (i.e. horizontally or vertically elongate) can have a differential effect on the sharpness of horizontal and vertical images [22], we did not distinguish between them because

vertical slit pupils are extremely uncommon among anurans and would compromise the statistical analyses.

Diel activity pattern is somewhat labile in anurans and can vary for specific behaviours; however, most species are predominantly nocturnal (solid lines in figure 1) [23], with only a few

lineages being predominantly diurnal, including the dendrobatoids (Aromobatidae + Dendrobatidae), hylodids, as well as *Atelopus* and *Brachycephalus* in our study (dashed lines in figure 1) [23]. We opted to handle diel pattern as a binary variable, in line with the previous work that uses this approach for different types of phylogenetic analyses [23]. Following this criterion, we scored *Scinax ruber* and *L. pipiens*, which have been reported to be arrhythmic [23], as nocturnal based on our own fieldwork experience.

Given that elevation and latitude contribute to shaping light habitats, we also took them into account. We scored the elevation and latitude of the same specimens used to obtain the lens transmittance measurements, with two caveats. First, *Bo. orientalis* was captive bred in the pet trade in Lund, Sweden (elevation: 51 metres above sea level (m.a.s.l.), latitude: 55.7°), which is within the natural elevation of the species but is approximately 7° north of its northernmost distribution [24]. As such, we performed analyses both including and excluding this species. Second, Kennedy & Milkman [7] did not provide collection data for the *L. pipiens* specimens they used to measure lens transmittance, but given that the research was conducted at Harvard University, which is within the natural distribution of the species [25], we assumed they were collected nearby.

## (c) Statistical analysis

We performed a phylogenetic comparative analysis to evaluate the linear relationships between lens $\lambda_{T50}$, eye size and pupil shape as predictor variables, and diel activity pattern, elevation and latitude as response variables. Given the reports of a linear relationship between lens $\lambda_{T50}$ and eye size in birds and mammals [5,8], we also tested this relationship explicitly. We used the phylogenetic hypothesis of Pyron [26] to control for the phylogenetic non-independence of the species in our sample (see the electronic supplementary material, S1E for details on how species missing in the tree were accommodated and the electronic supplementary material, S1F for the resulting tree).

We performed all analyses in R v. 3.6.0 using the packages *ape* v. 5.3 [27], *car* v. 3.0-3 [28], *GEIGER* v. 2.0.6.2 [29] and *nlme* v. 3.1-139 [30]. We used the function binaryPGLMM to run a phylogenetic generalized linear mixed model for binary data [31] for diel activity pattern. For the continuous response variables of elevation and latitude, we performed phylogenetic general least-squares analyses using the GLS function, correlation structures assuming a Brownian motion model of evolution and transformation of the variance–covariance matrix of the phylogeny using Pagel's $\lambda$ values of 0, 0.01, 0.5 and 1 [32]. We tested for multicollinearity using variance inflation factors.

## 3. Results

The lens $\lambda_{T50}$ of the 32 species measured in this study are spread throughout the UV–violet part of the spectrum, covering the range 280–425 nm (figure 1), which also contains the values of the five species in our previous study [21]. The breadth of the range is similar in Hyloides and Ranoides, the two major lineages of neobatrachians that contain more than 90% of the anuran species diversity [33], although the upper boundary for the latter seems to be lower than for the former (figure 1). The lowest value of the range ($\lambda_{T50} = 280$ nm) is that of *Bo. orientalis*, a comparatively basal species that is phylogenetically distant from the rest of the lineages in our sample (figure 1; electronic supplementary material, S1F).

We also measured the transmittance of the corneas in eight species from our sample. For most of them, the $\lambda_{T50}$ was within the range approximately 320–345 nm, irrespective of the lens

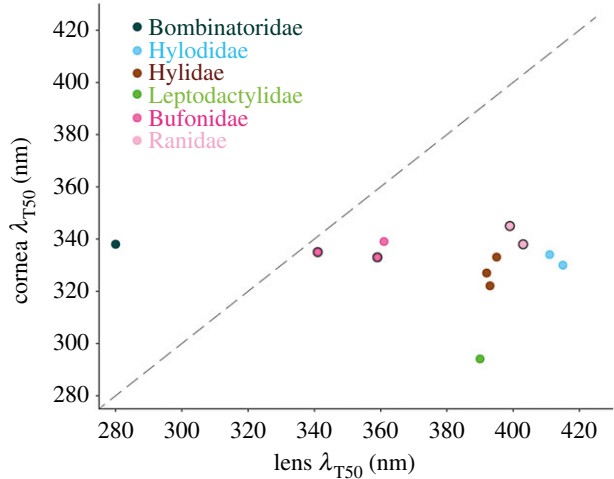

**Figure 2.** Relationship between the cornea and lens transmittance for a subset of the species used in the study. Each data point represents one species, and the data for the points with a black outline were obtained from a previous study [21]. The diagonal defines the regions in which light transmittance of the eye as a whole is limited by the cornea (above the line) and by the lens (under the line). See the electronic supplementary material, S1G for corneal transmittance curves, $\lambda_{T50}$ values and species identification, and the electronic supplementary material, S2 for full transmittance datasets. (Online version in colour.)

transmittance properties (figure 2; electronic supplementary material, S1G). However, *Physalaemus cuvieri* has a cornea $\lambda_{T50} = 293$ nm (figure 2), and that is probably also the case for *Xenohyla truncata*, although the precise $\lambda_{T50}$ value could not be calculated for the latter (electronic supplementary material, S1G).

The lens transmittance curves in our sample show the expected sigmoidal shape (figure 3; electronic supplementary material, S1H), with some variation in the slope of the short-wavelength cut-off and the saturation at long wavelengths. A notable feature in the shape of some of the curves is a localized increase in transmittance in the range approximately 310–340 nm (*Hylodes phyllodes*, *Oophaga pumilio*, *Dendropsophus microcephalus* and *Cochranella granulosa* in figure 3 and other species closely related to each of them; electronic supplementary material, S1H), as well as the shoulder in the same wavelength range in *Craugastor fitzingeri* and *Brachycephalus ephippium* (figure 3).

We found no correlation between lens $\lambda_{T50}$ and eye size, either controlling (Pagel's $\lambda = 1$; regression coefficient = 3.7164, $p = 0.1777$) or not controlling (Pagel's $\lambda = 0$; regression coefficient = 0.4643, $p = 0.868$; figure 4) for phylogeny.

We also found no significant relationship between lens $\lambda_{T50}$, eye size and pupil shape (predictor variables) and diel pattern, elevation and latitude (response variables) for any of the models ($p > 0.09$; see the electronic supplementary material, S1I).

## 4. Discussion

### (a) Limits of ultraviolet transmittance in anuran eyes

Our results show that lens transmittance among anurans spans a range similar to that of other vertebrates such as fishes, snakes, lizards, birds and mammals [5,6,8,16–19]. Our sample covers virtually the whole latitudinal range of geographical distribution of the group, a broad altitudinal

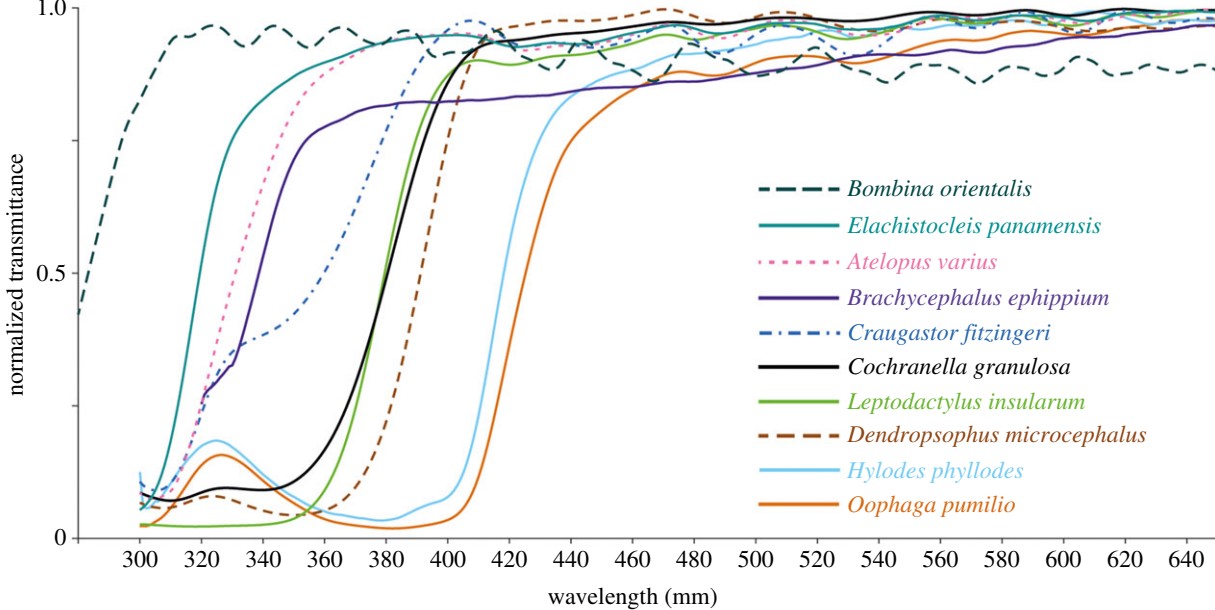

**Figure 3.** Lens transmittance curves from some of the species in the study. The *x*-axis is cut at 650 nm to ease visualization of the curves at short wavelengths. See the electronic supplementary material, S1H for curves from the rest of the species, and the electronic supplementary material, S2 for full transmittance datasets. (Online version in colour.)

range (0–1600 m.a.s.l.) and includes diurnal and nocturnal species (figure 1), thus covering the geographical and temporal environments to which most adult terrestrial anurans are exposed. It would be interesting to investigate whether the lenses of aquatic species and/or those that live in very high elevations show specific patterns within this range, or significant departures from it.

The lower limit in the range of lens $\lambda_{T50}$ among the anurans in our sample is intriguing, and more than 20 nm shorter than the most extreme cases reported so far in vertebrates: the porcupine fish *Diodon hystrix* (301 nm, [6]), the sand-dwelling lizard *Calyptommatus nicterus* (303 nm, [34]), the African house snake *Boaedon (Lamprophis) olivaceus* (306 nm, [17]) and the Japanese quail *Coturnix japonica* (approx. 310 nm, [35]). However, the unusually high lens UV transmission in *Bo. orientalis* has no functional relevance in terms of light availability for the retina, because the cornea of this frog has a $\lambda_{T50} = 338$ nm (similar to other species with higher lens $\lambda_{T50}$ values; figure 2; electronic supplementary material, S1G). This means that the amount of UV light that can effectively reach the photoreceptors is comparable to that in frogs with lens $\lambda_{T50} \approx 335–340$ nm. Thus, in this particular species, the light transmittance of the eye as a whole is limited by the cornea rather than the lens (figure 2), as is the case in quails [35]. However, this is probably the exception rather than the rule and not necessarily the case in other species in our sample with short lens $\lambda_{T50}$ values, such as *Ischnocnema parva*, *Engystomops pustulosus* or *Elachistocleis panamensis*. Even though we do not have data on the corneal transmittances of any of them, the data from other species in our sample (e.g. *Ph. cuvieri*) show that anuran corneas can in some cases transmit virtually all light down to 300 nm and further. Furthermore, a broad sample of fishes showed a presumptive trend for corneas to be more transmissive than lenses for any given species [16], although this relationship has not been formally tested for any vertebrate group.

The remarkably low lens $\lambda_{T50}$ that we found in *Bo. orientalis* is intriguing both from the point of view of lens

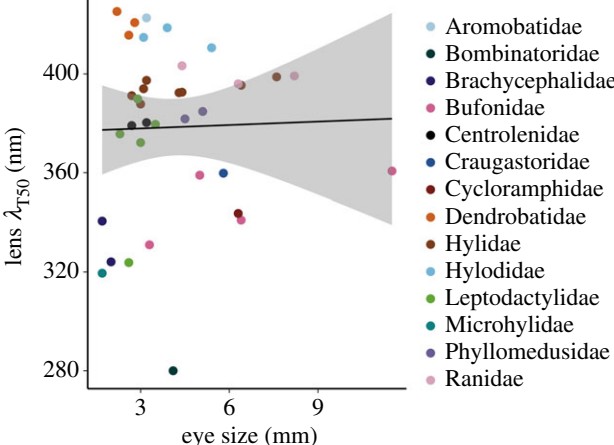

**Figure 4.** Relationship between lens transmittance and eye size. The shaded area is the 95% confidence interval. See the electronic supplementary material, S1 J for results obtained with a subset of species bearing putatively unpigmented lenses. (Online version in colour.)

structure and of the evolutionary history of UV transmittance in anurans. Biological tissues in general transmit only UV radiation greater than 310 nm, as aromatic amino acids absorb shorter wavelengths [1], so this lens might have either a specific spatial distribution of proteins, a low concentration of them, a particular crystalline type, or a combination of all these factors to achieve a $\lambda_{T50}$ value of 280 nm.

It is tempting to wonder whether this species is representative of the ancestral state of lens transmittance among anurans, given its basal phylogenetic position relative to the other species in our analyses and the fact that its cornea filters potentially damaging UV radiation, which might have relaxed selective pressure to make the lens less transmissive. Data from caudates—the sister group of anurans—seem to be limited to the salamander *Salamandra salamandra*, the newt *Cynops pyrrhogaster* and the axolotl *Ambystoma mexicanum* [1], all of which are deeply nested within Caudata [26].

Although lens $\lambda_{T50}$ values have not been published for any of them, a gross visual estimation from the available curves (in fig. 3b from [1]) suggests a range of 310–320 nm, which is intermediate between *Bombina* and the anuran species in our study. As our sample was too phylogenetically sparse to allow meaningful evaluation of the evolutionary history of this trait through ancestral state reconstruction, further studies focused on a richer sampling of anuran basal groups, as well as caudates, would be required to clarify this point.

## (b) How is ultraviolet filtering achieved in anuran lenses?

Variation in the shape of the transmittance curves among lenses that absorb part of the UV radiation is quite a common theme in vertebrates, and, in particular, local increases at short wavelengths can be seen in lens transmittance curves from mammals [5] and snakes [17]. However, the only group in which this phenomenon has been thoroughly studied are fishes, for which several pigments drive these patterns [6]: fishes with curves like the ones for *H. phyllodes* and *O. pumilio* have a pigment with peak absorption at approximately 370 nm, whereas other species of fishes with shoulders in their transmittance curves similar to those of *Br. ephippium* and *Cr. fitzingeri* have two different pigments with peak absorptions at approximately 320–330 and 360 nm. Finally, fish lenses with smooth curves and high $\lambda_{T50}$ values like the one from *Leptodactylus insularum* in our sample have high concentrations of either the 360 nm pigment or both the 320–330 and 360 nm pigment [6]. Curves with very subtle local increases at short wavelengths similar to those of *De. microcephalus* and *Co. granulosa* in our sample have not been reported in fishes, but are present in some mammals such as the meerkat, in whom lens pigments with absorption maxima at 360–370 nm have been extracted [5].

The similarity between fishes and anurans in the overall shape of transmittance curves for lenses of different species suggests that a number of pigments are involved in generating those patterns in the latter, as they are in the former. However, no comparative studies of lens pigmentation have been conducted in anurans. The only species for which a lens pigment has been extracted is the leopard frog *L. pipiens*; its absorbance peaks at 345 nm and it was not identified [7]. However, this absorbance profile does not match any other pigment identified in the lenses of fishes or mammals [1], so it is very likely that its chemical identity is different.

The presumptive presence of pigments in some anuran lenses can explain the lack of correlation between lens transmittance and eye size in our analyses, as that relationship holds only for unpigmented lenses [1]. It is thus possible that a relationship between the two variables exists in amphibians, as it does in birds [8,35], mammals [5] and some fishes [36], but is masked by the pigmented lenses in our sample. The absence of lens pigments has been demonstrated for 33 species of fishes with smooth transmittance curves and lens $\lambda_{T50} \approx 310$–340 nm [6]. Interestingly, if the linear regression for our sample is performed only with the six species that also have smooth transmittance curves and lens $\lambda_{T50} \approx 310$–340 nm, the relationship between lens transmittance and eye size has an excellent fit ($R^2 = 0.96$; electronic supplementary material, S1 J). If variation in the occurrence of pigment is confirmed for frog lenses, the relationship between lens transmittance and eye size should be retested among the species that fulfil the requirement of the absence of pigment.

## (c) Potential factors driving the relationship (or lack thereof) between lens transmittance and diel pattern

All studies that have qualitatively tested the hypothesis that lenses of diurnal and nocturnal vertebrates are UV-absorbing and UV-transmissive, respectively, mention deviations from this expected distribution pattern [5,15,17] that can seem anecdotal in each particular case, but taken together, they always point in the same direction: all nocturnal species have UV-transmissive lenses and all species with UV-absorbing lenses are diurnal, but some diurnal species have UV-transmissive lenses (figure 5). In this scenario, it comes as no surprise that there are diurnal anuran species in our sample on both sides of the UV transmission axis and no significant correlation of lens transmittance with diel pattern (and by extension, with the remaining variables that influence intensity and spectral composition of the light environment).

Despite their shared propensity to transmit at least part of the incoming UV radiation through their ocular media, the benefits might differ among nocturnal species from different vertebrate groups. Nocturnal vision in vertebrates is driven by rod photoreceptors, which typically have a peak spectral sensitivity outside the UV range at approximately 500 nm [37]. In addition, rods, as well as all other vertebrate photoreceptors, have a secondary lower, broader peak in the UV range (the β-band) [37] whose contribution to overall photon catch can become relevant and improve visual sensitivity in dim light when the total number of photons is extremely limited. In the case of amphibians, their rods are generally bigger—and thus more sensitive to wavelengths close to the spectral sensitivity peak—than those of other vertebrates [38], so the contribution of UV light to overall visual sensitivity might not be as crucial as in other groups. Indeed, the lenses of many nocturnal frogs are close to the boundary between UV-transmissive and UV-absorbing (e.g. some hylids and ranids, figure 5) and absorb almost all light in the region of the β-band [21]. However, anurans and some caudates are unique among vertebrates in having a second rod type with peak sensitivity at approximately 435 nm, in addition to the typical one at approximately 500 nm [37,39]. This dual rod system allows frogs to retain the ability to discriminate colours down to light intensities in which other vertebrates become colour-blind [40,41], and its proper functioning might be relevant for many of the approximately 80% of anuran species that are nocturnal [23]. In this context, it becomes crucial that the lens does not absorb too much short-wavelength light; $\lambda_{T50} = 403$ nm already reduces a significant amount of the light that can reach the retina in *Ra. temporaria* and removes almost completely the β-bands of both rods' spectral sensitivity curves [21]. Higher values could affect the sensitivity peak of the blue-sensitive rods, becoming detrimental to the performance of the visual system of nocturnal frogs in the dimly lit environments they inhabit.

As is the case with other vertebrates, there is no clear reason why some of the diurnal frogs in our sample depart from the expected UV-absorbing lenses. Filtering short-wavelength radiation can help reduce scattering and chromatic aberrations, thus improving spatial resolution, as has been suggested for

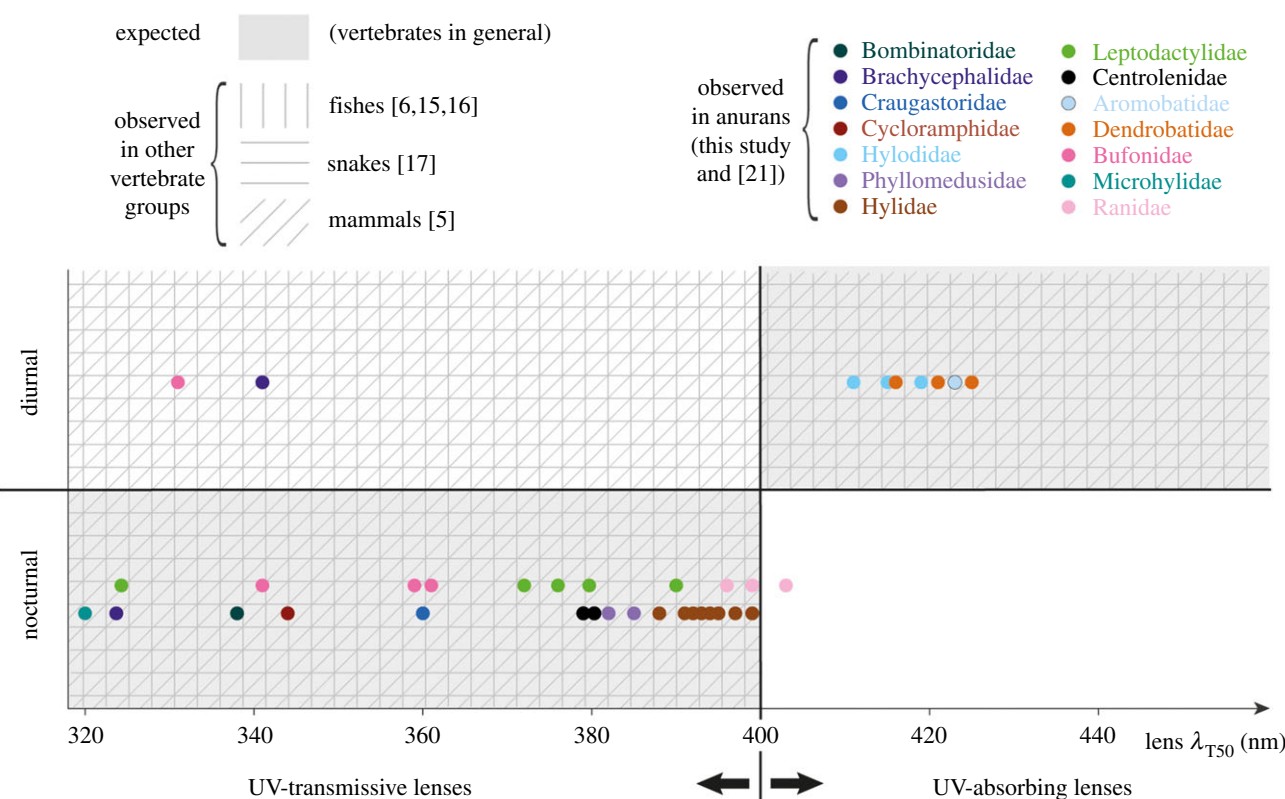

**Figure 5.** Distribution of lens transmittance values in diurnal and nocturnal representatives of different vertebrate groups. Each dot represents one anuran species. For Bombinatoridae (represented by *B. orientalis*), we show the cornea (rather than the lens) $\lambda_{T50}$ as it is the limiting component for UV transmission in this species. The boundary between UV transmission and absorption is an artificial barrier within a continuum and is meant to aid grouping and visualization. A compilation of exact lens $\lambda_{T50}$ values and diel patterns of species in groups other than anurans was outside the scope of this study, so those data are shown as general presence/absence patterns within each quadrant. (Online version in colour.)

animals that depend on sharp vision, such as raptors [8,20] and gliding snakes [17], and share UV-absorbing lenses. In the case of frogs, enhanced spatial resolution could be advantageous to species that use visual displays. All the diurnal representatives in our samples use them [42–45], suggesting that the optical problems caused by UV light are not serious enough—probably owing to the small size of their eyes—to drive the selective pressure towards longer-wavelength shifted lens $\lambda_{T50}$ values in all cases. An alternative explanation could be that, in some species, chromatic aberrations, if relevant, are dealt with by multifocal lenses rather than by UV-absorbing ones. Multifocality has only been tested in two anurans: the bufonid *Rhinella marina* (formerly *Bufo marinus*) has a multifocal lens, while the dendrobatid *Phyllobates bicolor* does not [10]. These data complement the differences in UV transmittance between the diurnal bufonid (*Atelopus*) and dendrobatids in our sample but are too limited to speculate about potential generalities. It would be interesting to obtain information about both lens transmittance and focal optics in the same species for a variety of anuran lineages, which would enable well-grounded hypotheses to be formulated about potential relationships between the two variables.

There is the possibility that UV light carries information valuable to some species in ways that are beyond both our knowledge of their visual ecology and our ability to imagine, given our own blindness in that part of the spectrum. In a recent study, it was shown that UV- and violet-light sensitivity can resolve habitat structure by increasing the contrast between the upper and lower surfaces of leaves to an extent that depends on the geometry of the canopy [46]. This previously unforeseen result showcases the way in which

subtleties obscured by broad temporal and spatial habitat classifications (e.g. diurnal, nocturnal, open or forest) can be the actual driving force underlying specific adaptations in traits that seem to deviate from expected patterns.

Finally, the 'mismatch' between lens transmittance and diel pattern in anurans in particular and in vertebrates in general might be related to phylogeny. For example, although we did not detect a significant phylogenetic effect in our data, it is evident that species of the same families tend to have similar lens transmittance properties, irrespective of whether they share the same diel pattern or not (e.g. Brachycephalidae, Bufonidae). This shows that within certain transmittance ranges and in the absence of highly specialized ecological demands, fluctuations in diel patterns within lineages have occurred without major departures from ancestral lens transmittance properties. The caveat that the phylogenetic constraints can be overridden by other factors is illustrated in our sample by the fact that the Túngara frog *Engystomops* (formerly *Physalaemus*) *pustulosus* has a lens $\lambda_{T50}$ value approximately 50 nm shorter than its close relative *Ph. cuvieri* and all other leptodactylids in our sample (figure 1). We hope that our work will encourage further research and data collection on ocular media transmittance from additional amphibian species to broaden our sampling, thus enabling robust testing of phylogenetic signals.

**Ethics.** The specimen of *Bombina orientalis* had been kept as a pet in Lund, Sweden, and was donated to us on the day of its decease. Specimens from Brazil were collected and euthanized under licences 13173-2 and 54599-3 from the Chico Mendes Institute for Biodiversity Conservation and Biodiversity Authorisation and Information System (ICMBio/SISBIO). The method for euthanasia was selected and applied according to Resolution no. 37 of the Brazilian National Council for Control of

Animal Experimentation. Specimens from Panama were collected under the permits SE/A-47-18, DAPB-0407-2019, SC/A-7-19, DAPB-0407-2019, SE/AP-8-19, issued by the Ministerio de Ambiente.

Data accessibility. Additional details about methods and results, as well as all transmittance datasets, are available in the electronic supplementary material.

Authors' contributions. C.A.M.Y., M.E.R.P., A.K. and T.G. conceptualized the study. C.A.M.Y., M.E.R.P., R.I. and T.G. conducted fieldwork. C.A.M.Y., M.E.R.P., G.J.-C. and R.I. collected data. C.A.M.Y., M.E.R.P., A.K. and T.G. analysed data. C.A.M.Y. wrote the manuscript with feedback from all authors. All authors approved the final version of the manuscript.

Competing interests. We have no competing interests.

Funding. This work was supported by Swedish Research Links (grant no. 2014-303-110535-69), São Paulo Research Foundation (grant nos 2015/14857-6, 2018/11502-0, 2012/10000-5, 2018/15425-0 and 2011/50146-6), the Brazilian National Council for Scientific and Technological Development (grant no. 306823/2017-9), the Sistema Nacional de Investigación de Panamá, the Panamá Amphibian Rescue and Conservation Project and Minera Panamá.

Acknowledgements. We are thankful to Miguel Trefaut Rodrigues for providing specimens and to Marco A. de Sena, José Mario Gellere, Hugo Bonfim, Isabela R. Cavalcanti, Sergio M. de Souza, Agustín Camacho Guerrero, Délio Baêta, Ariadne F. Sabbag, Carla M. Lopes, Jhon Jairo Ospina Sarria, Andrés Brunetti, Hélio R. da Silva, Edivaldo Vasconcelos de Farias, Pedro Henrique Salamão Gananga, Alfredo Pedroso dos Santos, Síria Ribeiro and Ricardo Cossio for help with logistics and fieldwork. Our grateful thanks also goes to Thais Condez, Ariadne F. Sabbag, Mariane Targino and Marco A. Rada García for generously sharing their data on eye sizes, and to the reviewers for their comments on the manuscript.

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
