## [Reviewer comments · Proceedings of the Royal Society B: Biological Sciences]

Review History

RSPB-2019-2253.R0 (Original submission)

Review form: Reviewer 1

Recommendation

Accept with minor revision (please list in comments)

Scientific importance: Is the manuscript an original and important contribution to its field?

Acceptable

General interest: Is the paper of sufficient general interest?

Acceptable

Quality of the paper: Is the overall quality of the paper suitable?

Good

Is the length of the paper justified?

Yes

Should the paper be seen by a specialist statistical reviewer?

No

Do you have any concerns about statistical analyses in this paper? If so, please specify them explicitly in your report.

No

It is a condition of publication that authors make their supporting data, code and materials available - either as supplementary material or hosted in an external repository. Please rate, if applicable, the supporting data on the following criteria.

Is it accessible?

Yes

Is it clear?

Yes

Is it adequate?

No

Do you have any ethical concerns with this paper?

No

Comments to the Author

- 1) Why was the transmittance of the corneas not measured in all species to get a better picture of overall ocular media transmittance? Too late now of course, but the reason should be mentioned.
- 2) The lack of any significant relationship between lambda T50 and elevation and latitude should also be mentioned in the abstract.
- 3) Line 40 - better: "...it absorbs light strongly below 310nm"
- 4) Line 40 - better: "Thus the longer the optical pathlength (anterior-posterior diameter) of the lens..."
- 5) Line 229 - Figure 4 legend - reword? The stats say no correlation between eye size and lambda T50 so perhaps the figure should not be described as a Linear relationship?
- 6) Line 227 space between "amino" and "acids"
- 7) Line 357 - I am not sure this necessarily follows - rods may be bigger but it is overall photon catch that is important and there is nothing to suggest that they would selectively block UV wavelengths if they were indeed useful, especially given the presumed metabolic costs of maintaining larger rod outer segments. Remove or rephrase?

Review form: Reviewer 2

Recommendation

Accept with minor revision (please list in comments)

Scientific importance: Is the manuscript an original and important contribution to its field?

Good

General interest: Is the paper of sufficient general interest?

Good

Quality of the paper: Is the overall quality of the paper suitable?

Excellent

Is the length of the paper justified?

Yes

Should the paper be seen by a specialist statistical reviewer?

No

Do you have any concerns about statistical analyses in this paper? If so, please specify them explicitly in your report.

No

It is a condition of publication that authors make their supporting data, code and materials available - either as supplementary material or hosted in an external repository. Please rate, if applicable, the supporting data on the following criteria.

Is it accessible?

N/A

Is it clear?

N/A

Is it adequate?

N/A

Do you have any ethical concerns with this paper?

No

Comments to the Author

This is a generally well written and well presented manuscript and data-set that will add to our understanding of vision in frogs and indeed vertebrates in general.

I have only minor comments and a couple of recommendations for future studies.

Starting with the recommendations.

a) Given the access to fresh tissue, rather than poke out a lens and measure it and then make up excuses around the number of corneas measured etc, it would be / would have been better to measure whole-eye transmittance. This can be done relatively easily by cutting a window in the back of the eye, washing and then using the globe as a whole. This gives the best measure of the light that actually reaches the retina and is most relevant for both the phylogenetic and ecological correlations you wish to draw. Of course, the individual components can then be measured afterwards for other purposes.

b) If you want to make comment / correlate ocular media transmissions with light environment, it is best to measure these and comments around elevation and latitude etc are not very relevant -

especially in the case of the previously captive frog. If there is variation there it is more likely due to the likely indoor and totally unnatural light environment it was kept in. While it is OK to retain the elevation and altitude as a data-set to seek correlation with, the use of this as a proxy for light environment (or implication of this) should be removed. You may find it worthwhile to go to the frogs actual habitats and measure the light available to them using would Ocean Optics spec.

Small things per line:

21 do you mean nocturnal not basal?

21 "... and a cut-off range that..."

22 maybe define as human visible spectrum later on. In general visible spectrum is used to mean 400-700 for us lot but always good to define with a sentence somewhere.

28 "...that anuran ocular media..."

36-39 this is a strange way to say what you are saying and is confusing - why not just say the cut-off is variable?

40 "Also, the bigger the..."

45 "...and better spatial..."

49 "Prolonged exposition..." too many Furthermores!!

50 "...may also cause photochemical..."

55-64 seems out of place and could be moved to discussion for clarity?

86 "... eyes remains in its infancy." On this - remember doing a statistical test on a correlation does not necessarily make it more valid scientifically - stats is a support not an end-point.

88 "... differ by more than..."

90 "... , broader sampling was clearly needed..."

110 "...samples were measured..."

127 please define what you mean by 'cut for clarity'

245 remove the idea that you have 'covered the diversity of light', until you have actually measured this in the habitat / microhabitat for each species you have not done this.

346 Same as comment for 245, remove the idea of 'and by extension...' you need to measure the light properly, not rely on a proxy.

365 It is not crucial that a lens does not absorb below 400 for the potential for colour vision between rods as inferred here. It is less good for overall sensitivity and the B-band absorbance of photons but not for potential rod dichromacy.

On this point - you do not discuss anywhere the cones that many frogs have and the data there? It would be worth including a brief review-table of this as known for the species you have. There will be many holes in this data but still worth a comment and the potential for further correlative arm-waving.

Decision letter (RSPB-2019-2253.R0)

15-Nov-2019

Dear Dr Yovanovich

I am pleased to inform you that your Review manuscript RSPB-2019-2253 entitled "Lens transmittance shapes UV sensitivity in the eyes of frogs from diverse ecological and phylogenetic backgrounds" has been accepted for publication in Proceedings B.

The referee(s) do not recommend any further changes. Therefore, please proof-read your manuscript carefully and upload your final files for publication. Because the schedule for publication is very tight, it is a condition of publication that you submit the revised version of your manuscript within 7 days. If you do not think you will be able to meet this date please let me know immediately.

To upload your manuscript, log into <http://mc.manuscriptcentral.com/prsb> and enter your Author Centre, where you will find your manuscript title listed under "Manuscripts with Decisions." Under "Actions," click on "Create a Revision." Your manuscript number has been appended to denote a revision.

You will be unable to make your revisions on the originally submitted version of the manuscript. Instead, upload a new version through your Author Centre.

1) A text file of the manuscript (doc, txt, rtf or tex), including the references, tables (including captions) and figure captions. Please remove any tracked changes from the text before submission. PDF files are not an accepted format for the "Main Document".

2) A separate electronic file of each figure (tiff, EPS or print-quality PDF preferred). The format should be produced directly from original creation package, or original software format. Please note that PowerPoint files are not accepted.

3) Electronic supplementary material: this should be contained in a separate file from the main text and the file name should contain the author's name and journal name, e.g. `authorname_procb_ESM_figures.pdf`

All supplementary materials accompanying an accepted article will be treated as in their final form. They will be published alongside the paper on the journal website and posted on the online figshare repository. Files on figshare will be made available approximately one week before the accompanying article so that the supplementary material can be attributed a unique DOI. Please see: <https://royalsociety.org/journals/authors/author-guidelines/>

4) Data-Sharing and data citation

It is a condition of publication that data supporting your paper are made available. Data should be made available either in the electronic supplementary material or through an appropriate repository. Details of how to access data should be included in your paper. Please see <https://royalsociety.org/journals/ethics-policies/data-sharing-mining/> for more details.

<http://datadryad.org/submit?journalID=RSPB&manu=RSPB-2019-2253> which will take you to your unique entry in the Dryad repository.

Once again, thank you for submitting your manuscript to Proceedings B and I look forward to

receiving your final version. If you have any questions at all, please do not hesitate to get in touch.

Sincerely,
Dr Sasha Dall
mailto:proceedingsb@royalsociety.org

Associate Editor Board Member: 1

Comments to Author:

Two reviewers reviewed this manuscript. In both cases, the reviewers demonstrated support for the work, with overall acceptable to good support for scientific importance and general interest, as well as good to excellence regarding the quality of the paper. Generally the work is not novel per se, but adds much useful data to the field of optics and visual ecology. In general, the reviewers commented that the paper would suit a broad audience (e.g. PRSB).

Some relatively minor comments were made which should be addressed before a final decision may be made. I highlight some of the important issues below:

1. Full raw spectral transmittance data for the ocular media should be provided - not just the normalised data.
2. Please strongly consider the recommendations (a) and (b) as suggested by Rev 2.
3. Please include any data on frog cones, or at least discuss cones where appropriate.

Kind regards.

Reviewer(s)' Comments to Author:

Referee: 1

Comments to the Author(s)

- 1) Why was the transmittance of the corneas not measured in all species to get a better picture of overall ocular media transmittance? Too late now of course, but the reason should be mentioned.
- 2) The lack of any significant relationship between lambda T50 and elevation and latitude should also be mentioned in the abstract.
- 3) Line 40 - better: "...it absorbs light strongly below 310nm"
- 4) Line 40 - better: "Thus the longer the optical pathlength (anterior-posterior diameter) of the lens..."
- 5) Line 229 - Figure 4 legend - reword? The stats say no correlation between eye size and lambda T50 so perhaps the figure should not be described as a Linear relationship?
- 6) Line 227 space between "amino" and "acids"
- 7) Line 357 - I am not sure this necessarily follows - rods may be bigger but it is overall photon catch that is important and there is nothing to suggest that they would selectively block UV wavelengths if they were indeed useful, especially given the presumed metabolic costs of maintaining larger rod outer segments. Remove or rephrase?

Referee: 2

Comments to the Author(s)

This is a generally well written and well presented manuscript and data-set that will add to our understanding of vision in frogs and indeed vertebrates in general.

I have only minor comments and a couple of recommendations for future studies.

Starting with the recommendations.

a) Given the access to fresh tissue, rather than poke out a lens and measure it and then make up excuses around the number of corneas measured etc, it would be / would have been better to measure whole-eye transmittance. This can be done relatively easily by cutting a window in the back of the eye, washing and then using the globe as a whole. This gives the best measure of the light that actually reaches the retina and is most relevant for both the phylogenetic and ecological correlations you wish to draw. Of course, the individual components can then be measured afterwards for other purposes.

b) If you want to make comment / correlate ocular media transmissions with light environment, it is best to measure these and comments around elevation and latitude etc are not very relevant - especially in the case of the previously captive frog. If there is variation there it is more likely due to the likely indoor and totally unnatural light environment it was kept in. While it is OK to retain the elevation and altitude as a data-set to seek correlation with, the use of this as a proxy for light environment (or implication of this) should be removed. You may find it worthwhile to go to the frogs actual habitats and measure the light available to them using Ocean Optics spec.

Small things per line:

21 do you mean nocturnal not basal?

21 "... and a cut-off range that..."

22 maybe define as human visible spectrum later on. In general visible spectrum is used to mean 400-700 for us lot but always good to define with a sentence somewhere.

28 "...that anuran ocular media..."

36-39 this is a strange way to say what you are saying and is confusing - why not just say the cut-off is variable?

40 "Also, the bigger the..."

45 "...and better spatial..."

49 "Prolonged exposition..." too many Furthermores!!

50 "...may also cause photochemical..."

55-64 seems out of place and could be moved to discussion for clarity?

86 "... eyes remains in its infancy." On this - remember doing a statistical test on a correlation does not necessarily make it more valid scientifically - stats is a support not an end-point.

88 "... differ by more than..."

90 "... , broader sampling was clearly needed..."

110 "...samples were measured..."

127 please define what you mean by 'cut for clarity'

245 remove the idea that you have 'covered the diversity of light', until you have actually measured this in the habitat / microhabitat for each species you have not done this.

346 Same as comment for 245, remove the idea of 'and by extension...' you need to measure the light properly, not rely on a proxy.

365 It is not crucial that a lens does not absorb below 400 for the potential for colour vision between rods as inferred here. It is less good for overall sensitivity and the B-band absorbance of photons but not for potential rod dichromacy.

On this point - you do not discuss anywhere the cones that many frogs have and the data there? It would be worth including a brief review-table of this as known for the species you have. There will be many holes in this data but still worth a comment and the potential for further correlative arm-waving.

Author's Response to Decision Letter for (RSPB-2019-2253.R0)

See Appendix A.

Decision letter (RSPB-2019-2253.R1)

02-Dec-2019

Dear Dr Yovanovich

I am pleased to inform you that your manuscript entitled "Lens transmittance shapes UV sensitivity in the eyes of frogs from diverse ecological and phylogenetic backgrounds" has been accepted for publication in Proceedings B.

Open Access

You are invited to opt for Open Access, making your freely available to all as soon as it is ready for publication under a CCBY licence. Our article processing charge for Open Access is £1700. Corresponding authors from member institutions (<http://royalsocietypublishing.org/site/librarians/allmembers.xhtml>) receive a 25% discount to these charges. For more information please visit <http://royalsocietypublishing.org/open-access>.

Paper charges

Sincerely,

Dr Sasha Dall
Editor, Proceedings B
mailto: proceedingsb@royalsociety.org

Appendix A

Yovanovich *et al.* – RESPONSE TO REVIEWERS

Referee: 1

Comments to the Author(s)

1) *Why was the transmittance of the corneas not measured in all species to get a better picture of overall ocular media transmittance? Too late now of course, but the reason should be mentioned.*

Our study was focused on lens transmittance given the greater amount of literature available for comparison and discussion, and the higher variability previously documented in other groups. Depending on the species, both the lenses and corneas we used can degrade quite fast once dissected, so we always prioritised measuring the lenses in the first place. Furthermore, we also needed to keep the eyecups as intact as possible for other projects. Thus, corneas were not always available/well preserved enough for transmittance measurements and should be regarded as a 'bonus' that was included in the manuscript because it is information that can be useful about the species for which we collected it and deserves to be made available, even if the dataset doesn't cover our whole sample.

2) *The lack of any significant relationship between lambda T50 and elevation and latitude should also be mentioned in the abstract.*

Thanks for the suggestion. The abstract is just one word short of the maximum length allowed by the journal, so it is not possible to add more detail to it.

3) *Line 40 - better: "...it absorbs light strongly below 310nm"*

Corrected.

4) *Line 40 - better: "Thus the longer the optical pathlength (anterior-posterior diameter) of the lens..."*

Thanks for the suggestion; given that the content of the sentence would remain the same we have opted to keep the original phrasing.

5) *Line 229 - Figure 4 legend - reword? The stats say no correlation between eye size and lambda T50 so perhaps the figure should not be described as a Linear relationship?*

[Line 556] We have removed the word "Linear" from the figure legend.

6) *Line 227 space between "amino" and "acids"*

[Line 250] Corrected.

7) *Line 357 - I am not sure this necessarily follows - rods may be bigger but it is overall photon catch*

that is important and there is nothing to suggest that they would selectively block UV wavelengths if they were indeed useful, especially given the presumed metabolic costs of maintaining larger rod outer segments. Remove or rephrase?

[Line 325] We have edited this sentence to make it clearer.

Referee: 2

Comments to the Author(s)

This is a generally well written and well presented manuscript and data-set that will add to our understanding of vision in frogs and indeed vertebrates in general.

I have only minor comments and a couple of recommendations for future studies.

Starting with the recommendations.

a) Given the access to fresh tissue, rather than poke out a lens and measure it and then make up excuses around the number of corneas measured etc, it would be / would have been better to measure whole-eye transmittance. This can be done relatively easily by cutting a window in the back of the eye, washing and then using the globe as a whole. This gives the best measure of the light that actually reaches the retina and is most relevant for both the phylogenetic and ecological correlations you wish to draw. Of course, the individual components can then be measured afterwards for other purposes.

This strategy is certainly a valid option, and we agree that it would be more representative of what happens in an intact eye. However, we used the retinas from the same animals for other projects so we needed to keep the eyecups as intact as possible. Thus we deliberately chose to use the approach described in the manuscript to make the most out of the animals that we sacrificed.

b) If you want to make comment / correlate ocular media transmissions with light environment, it is best to measure these and comments around elevation and latitude etc are not very relevant - especially in the case of the previously captive frog. If there is variation there it is more likely due to the likely indoor and totally unnatural light environment it was kept in. While it is OK to retain the elevation and altitude as a data-set to seek correlation with, the use of this as a proxy for light environment (or implication of this) should be removed. You may find it worthwhile to go to the frogs actual habitats and measure the light available to them using Ocean Optics spec.

Please note that we have taken the particular situation of the captive frog into account by running our analyses both including and excluding that species, as we mention in lines 159-160. As for the use of elevation and latitudes, we agree that they don't replace spectral measurements in situ, which were outside the scope of our study. As stated by the reviewer, we have used them as a dataset for correlation analyses, acknowledging that they influence the light environment, but without implying that they fully represent it.

Small things per line:

21 do you mean nocturnal not basal?

We mean basal indeed.

21 "... and a cut-off range that..."

Corrected.

22 maybe define as human visible spectrum later on. In general visible spectrum is used to mean 400-700 for us lot but always good to define with a sentence somewhere.

Thanks for the suggestion. We have used specific wavelength ranges throughout the rest of the text rather than the expression "visible spectrum".

28 "...that anuran ocular media..."

Corrected.

36-39 this is a strange way to say what you are saying and is confusing - why not just say the cut-off is variable?

We have decided to keep the original wording, as it contains the information that we want to deliver (even if it is not phrased in the most commonly used way).

40 "Also, the bigger the..."

Thanks for the suggestion. We have decided to keep "Thus" to emphasize that the effect of size is a consequence of the amount of absorbing material inside the lens.

45 "...and better spatial..."

Corrected.

49 "Prolonged exposition..." too many Furthermores!! & 50 "...may also cause photochemical..."

Corrected.

55-64 seems out of place and could be moved to discussion for clarity?

We have kept this fragment in its original location to preserve the logical flow of the Introduction, as it contains the rationale for including pupil shapes in the study.

86 "... eyes remains in its infancy." On this - remember doing a statistical test on a correlation does not necessarily make it more valid scientifically - stats is a support not an end-point.

We fully agree with this view about what stats can and cannot do!

88 "*.. differ by more than..*"

Corrected.

90 "*.. , broader sampling was clearly needed..*"

Corrected.

110 "*..samples were measured..*"

[Line 111] Corrected.

127 *please define what you mean by 'cut for clarity'*

We have rephrased this sentence to make it clearer.

245 *remove the idea that you have 'covered the diversity of light', until you have actually measured this in the habitat / microhabitat for each species you have not done this.*

[Line 223] We have rephrased this sentence to make it clear that we refer to the geographical determinants of light environment, rather than the ones specified in your habitat/microhabitat.

346 *Same as comment for 245, remove the idea of 'and by extension...' you need to measure the light properly, not rely on a proxy.*

[Line 314] Please note that the sentence doesn't say that there is no correlation between transmittance and light intensity/spectral composition. It just comments on the absence of significant correlation between transmittance and the geographical variables that we tested in our analyses.

365 *It is not crucial that a lens does not absorb below 400 for the potential for colour vision between rods as inferred here. It is less good for overall sensitivity and the B-band absorbance of photons but not for potential rod dichromacy.*

[Line 335] Please note that this sentence refers to "short wavelength light" broadly, and it discusses how significant absorbance of light >400 nm (not <400 nm) would be detrimental for rod dichromacy.

On this point - you do not discuss anywhere the cones that many frogs have and the data there? It would be worth including a brief review-table of this as known for the species you have. There will be many holes in this data but still worth a comment and the potential for further correlative arm-waving.

We have considered discussing cones when preparing the manuscript, and decided against it to prioritise other topics more directly related to the variables that we worked it, and for which there is more information available allowing for deeper discussion, while keeping the length of the manuscript within the limits of the journal. Like many other features of the visual system, cones are severely understudied from the point of view of anuran diversity. To the best of our knowledge, the only lineages for which there is information available about cone spectral types are *Rana*, *Lithobates* and *Oophaga*, so a table like the one suggested here would have way too many holes at the moment. We hope that our work will elicit further interest in the visual systems of different frog species such that said table becomes feasible and benefits from the data that we provide here.